# Coronavirus Diseases (COVID-19) Current Status and Future Perspectives: A Narrative Review

**DOI:** 10.3390/ijerph17082690

**Published:** 2020-04-14

**Authors:** Francesco Di Gennaro, Damiano Pizzol, Claudia Marotta, Mario Antunes, Vincenzo Racalbuto, Nicola Veronese, Lee Smith

**Affiliations:** 1IRCCS Istituto Neurologico Mediterraneo NEUROMED, 86077 Pozzilli, Italy; cicciodigennaro@yahoo.it (F.D.G.); marotta.claudia@gmail.com (C.M.); 2Italian Agency for Development Cooperation, Khartoum 79371, Sudan; vincenzo.racalbuto@aics.gov.it; 3Department of Surgery, Central Hospital of Beira, Beira 2102, Mozambique; majomantu@gmail.com; 4Geriatric Unit, Department of Internal Medicine and Geriatrics, University of Palermo, 90100 Palermo, Italy; ilmannato@gmail.com; 5The Cambridge Centre for Sport and Exercise Sciences, Anglia Ruskin University, Cambridge CB1 1PT, UK; Lee.Smith@anglia.ac.uk

**Keywords:** coronavirus, COVID-19, pathogenesis, preparedness, emergency, pandemic

## Abstract

At the end of 2019 a novel virus, severe acute respiratory syndrome coronavirus 2 (SARS-CoV-2), causing severe acute respiratory syndrome expanded globally from Wuhan, China. In March 2020 the World Health Organization declared the SARS-Cov-2 virus a global pandemic. We performed a narrative review to describe existing literature with regard to Corona Virus Disease 2019 (COVID-19) epidemiology, pathophysiology, diagnosis, management and future perspective. MEDLINE, EMBASE and Scopus databases were searched for relevant articles. Although only when the pandemic ends it will be possible to assess the full health, social and economic impact of this global disaster, this review represents a picture of the current state of the art. In particular, we focus on public health impact, pathophysiology and clinical manifestations, diagnosis, case management, emergency response and preparedness.

## 1. Introduction

At the end of 2019, a series of pneumonia cases of unknown cause emerged in Wuhan (Hubei, China) [1]. A few weeks later, in January 2020, deep sequencing analysis from lower respiratory tract samples identified a novel virus severe acute respiratory syndrome coronavirus 2 (SARS-CoV-2) as causative agent for that observed pneumonia cluster [2]. On February 11th, 2020, the World Health Organization (WHO) Director-General, Dr. Tedros Adhanom Ghebreyesus, named the disease caused by the SARS-CoV-2 as “COVID-19”, and by March 11th, 2020 when the number of countries involved was 114, with more than 118,000 cases and over 4000 deaths, the WHO declared the pandemic status [3].

Corona Virus Disease 2019 (COVID-19) is an RNA virus, with a typical crown-like appearance under an electron microscope due to the presence of glycoprotein spikes on its envelope [4]. It is not the first time that a coronavirus causing an epidemic has been a significant global health threat: in November 2019, an outbreak of coronaviruses (CoVs) with severe acute respiratory syndrome (SARS)-CoV started in the Chinese province of Guangdong and again, in September 2012 the Middle East respiratory syndrome (MERS)-Co V appeared [5]. There are four genera of CoVs: (I) α-coronavirus (alphaCoV), (II) β-coronavirus (betaCoV) probably present in bats and rodents, while (III) δ-coronavirus (deltaCoV), and (IV) γ-coronavirus (gammaCoV) probably represent avian species [4,5,6]. The virus has a natural and zoonotic origin: two scenarios that can plausibly explain the origin of SARS-CoV2 are: (i) natural selection in an animal host before zoonotic transfer; and (ii) natural selection in humans following zoonotic transfer [5,6]. Clinical features and risk factors are highly variable, making the clinical severity range from asymptomatic to fatal [7]. Understanding of COVID-19 is on-going. This review aims to summarize early findings on the epidemiology, clinical features, diagnosis, management, and prevention of COVID-19.

### 1.1. Epidemiology

The COVID-19 epidemic expanded in early December from Wuhan, China’s 7th most populous city, throughout China and was then exported to a growing number of countries. The first confirmed case of COVID-19 outside China was diagnosed on 13th January 2020 in Bangkok (Thailand) [8]. On the 2nd of March 2020, 67 territories outside mainland China had reported 8565 confirmed cases of COVID-19 with 132 deaths, as well as significant community transmission occurring in several countries worldwide, including Iran and Italy and it was declared a global pandemic by the WHO on the 11th of March 2020 [9]. The number of confirmed cases is constantly increasing worldwide and after Asian and European regions, a steep increase in cases is currently (31 March 2020) being observed in low-income countries [10]. It is problematic to quantify the exact size of this pandemia as it would necessary to count all cases including not only severe and symptomatic cases but also mild ones [11]. Unfortunately, to date, there is not a global and standard response to the pandemia and each country is facing the crisis based on their own possibilities, expertise and hypotheses. Thus, there are different criteria for testing, hospitalisation and estimating of cases making it difficult to calculate the number of people affected by epidemic. Based on the data we have so far, the estimated case fatality ratio among medically attended patients is approximately 2%, but, also in this case, a true ratio may not be known for some time [12].

Today, 31st of March 2020, based on the WHO reports, we have globally 693,224 confirmed cases and 33,106 deaths, distributed as follows: Western Pacific Region 103,775 cases and 3649 deaths, European Region 392,757 cases and 29,962 deaths, South East Asia Region 4084 cases and 158 deaths, Eastern Mediterranean Region 46,329 cases and 2813 deaths, Region of the Americas 142,081 cases and 2457 deaths and in the Africa region 3486 cases and 60 deaths [13].

### 1.2. Pathophysiology and Clinical Manifestation

To address the pathogenetic mechanisms of SARS-CoV-2, its viral structure and genome must be considered. Coronaviruses are enveloped positive strand RNA viruses with the largest known RNA genomes—30–32 kb—with a 5′-cap structure and 3′-poly-A tail. Starting from the viral RNA, the synthesis of polyprotein 1a/1ab (pp1a/pp1ab) in the host is realized [14]. The transcription works through the replication-transcription complex (RCT) organized in double-membrane vesicles and via the synthesis of subgenomic RNAs (sgRNAs) sequences. Of note, transcription termination occurs at transcription regulatory sequences, located between the so-called open reading frames (ORFs) that work as templates for the production of subgenomic mRNAs [15]. In the atypical CoV genome, at least six ORFs can be present. Among these, a frameshift between ORF1a and ORF1b guides the production of both pp1a and pp1ab polypeptides that are processed by virally encoded chymotrypsin-like protease (3CLpro) or main protease (Mpro), as well as one or two papain-like proteases for producing 16 non-structural proteins (nsps) [15]. Apart from ORF1a and ORF1b, other ORFs encode for structural proteins, including spike, membrane, envelope, and nucleocapsid proteins and accessory proteic chains [14,15]. Different CoVs present special structural and accessory proteins translated by dedicated sgRNAs. Pathophysiology and virulence mechanisms of CoVs, and therefore also of SARS-CoV-2 have links to the function of the nsps and structural proteins. For instance, research has underlined that nsps are able to block the host innate immune response [16]. Among the functions of the structural proteins, the envelope has a crucial role in virus pathogenicity as it promotes viral assembly and release.

The pathogenic mechanism that produces pneumonia seems to be particularly complex [14,15,16]. The data so far available seem to indicate that the viral infection is capable of producing an excessive immune reaction in the host. In some cases, a reaction takes place, which as a whole is labelled a “cytokine storm”. The effect is extensive tissue damage. The protagonist of this storm is interleukin 6 (IL-6). IL-6 is produced by activated leukocytes and acts on a large number of cells and tissues [17]. It is able to promote the differentiation of B lymphocytes, promotes the growth of some categories of cells, and inhibits the growth of others. It also stimulates the production of acute phase proteins and plays an important role in thermoregulation, in bone maintenance and in the functionality of the central nervous system [18]. Although the main role played by IL-6 is pro-inflammatory, it can also have anti-inflammatory effects. In turn, IL-6 increases during inflammatory diseases, infections, autoimmune disorders, cardiovascular diseases and some types of cancer [19]. It is also implicated into the pathogenesis of the cytokine release syndrome (CRS) that is an acute systemic inflammatory syndrome characterized by fever and multiple organ dysfunction [20].

The virus might pass through the mucous membranes, especially nasal and larynx mucosa, then enters the lungs through the respiratory tract. Then the virus would attack the targeting organs that express angiotensin converting enzyme 2 (ACE2), such as the lungs, heart, renal system and gastrointestinal tract [18,19,20]. The virus begins a second attack, causing the patient’s condition to aggravate around 7 to 14 days after onset. B lymphocyte reduction may occur early in the disease, which may affect antibody production in the patient. Besides, the inflammatory factors associated with diseases mainly containing IL-6 were significantly increased, which also contributed to the aggravation of the disease around 2 to 10 days after onset.

The clinical spectrum of COVID-19 varies from asymptomatic or paucisymptomatic forms to clinical conditions characterized by severe respiratory failure that necessitates mechanical ventilation and support in an intensive care unit (ICU), to multiorgan and systemic manifestations in terms of sepsis, septic shock, and multiple organ dysfunction syndromes (MODS) [21]. Asymptomatic infections have also been described, but their frequency is unknown. The main symptoms are reported in Table 1. Pneumonia appears to be the most frequent serious manifestation of infection, characterized primarily by fever, cough, dyspnea, and bilateral infiltrates on chest imaging. [22] There are no specific clinical features that can yet reliably distinguish COVID-19 from other viral respiratory infections. Other, less common symptoms have included headaches, sore throat, and rhinorrhea. In addition to respiratory symptoms, gastrointestinal symptoms (e.g., nausea and diarrhea) have also been reported, and in some patients they may be the presenting complaint. Respiratory droplet transmission is the main route and it can also be transmitted through person-to-person contacts by asymptomatic carriers [21,22].

Chest CT in patients with COVID-19 most commonly demonstrates ground-glass opacification with or without consolidative abnormalities, consistent with viral pneumonia [23]. Chest CT abnormalities are more likely to be bilateral, have a peripheral distribution, and involve the lower lobes. Less common findings include pleural thickening, pleural effusion, and lymphadenopathy [24]. Chest CT may be helpful in making the diagnosis, but no finding can completely rule in or rule out the possibility of COVID-19. The possibility of COVID-19 should be considered primarily in patients with new onset fever and/or respiratory tract symptoms (e.g., cough, dyspnea) [23,24,25]. It should also be considered in patients with severe lower respiratory tract illness without any clear cause. Although these syndromes can occur with other viral respiratory illnesses, the likelihood of COVID-19 is increased if the patient [26]: (1) resides in or has travelled within the prior 14 days to a location where there is community transmission of SARS-CoV-2 (i.e., large numbers of cases that cannot be linked to specific transmission chains); (2) has had close contact with a confirmed or suspected case of COVID-19 in the prior 14 days, including through work in health care settings. Close contact includes being within approximately six feet (about two meters) of a patient for a prolonged period of time while not wearing personal protective equipment or having direct contact with infectious secretions while not wearing personal protective equipment.

The period from the onset of COVID-19 symptoms to death ranges from 6 to 41 days with a median of 14 days [27]. This period is dependent on the age of the patient and status of the patient’s immune system. It was shorter among patients >70-years old compared with those under the age of 70 years. The most common symptoms at onset of COVID-19 illness are fever, cough, and fatigue, while other symptoms include sputum production, headache, haemoptysis, diarrhoea, dyspnoea, and lymphopenia [26].

The WHO has reported an incubation period for COVID-19 between 2 and 10 days. However, some literature suggests that the incubation period can last longer than two weeks and it is possible that a very long incubation period could reflect double exposure [26]. Many studies support a 14-day medical observation period for people exposed to the pathogen. The severity of the clinical picture seems to be correlated with age (>70 years), comorbidities such as: diabetes, chronic obstructive pulmonary disease (COPD), hypertension, obesity and male sex but currently no scientifically valid explanations have been developed [27,28,29].

## 2. Diagnosis

For patients with suspected infection, the following diagnosis techniques are utilised: performing real-time fluorescence (RT-PCR) to detect the positive nucleic acid of SARS-CoV-2 in sputum, throat swabs, and secretions of the lower respiratory tract samples [30]. In patients with COVID-19, the white blood cell count can vary. Leukopenia, leukocytosis, and lymphopenia have been reported, although lymphopenia appears most common [30,31]. Elevated lactate dehydrogenase and ferritin levels are common, and elevated aminotransferase levels have also been described. On admission, many patients with pneumonia have normal serum procalcitonin levels; however, in those requiring ICU care, they are more likely to be elevated. High D-dimer levels and more severe lymphopenia have been associated with mortality. Imaging findings—Chest computed tomography (CT) in patients with COVID-19 most commonly demonstrates ground-glass opacification with or without consolidative abnormalities, consistent with viral pneumonia. Others study have suggested that chest CT abnormalities are more likely to be bilateral, have a peripheral distribution, and involve the lower lobes. Less common findings include pleural thickening, pleural effusion, and lymphadenopathy [23,24,25]. Chest CT may be helpful in making the diagnosis, but no finding can completely rule in or rule out the possibility of COVID-19.

An oropharyngeal swab can be collected but is not essential; if collected, it should be placed in the same container as the nasopharyngeal specimen. An oropharyngeal swab is an acceptable alternative if nasopharyngeal swabs are unavailable [32]. Expectorated sputum should be collected from patients with productive cough; induction of sputum is not recommended. A lower respiratory tract aspirate or bronchoalveolar lavage should be collected from patients who are intubated. Data from this study suggested that viral RNA levels are higher and more frequently detected in nasal compared with oral specimens, although only eight nasal swabs were tested. SARS-CoV-2 RNA is detected by reverse-transcription polymerase chain reaction (RT-PCR) [33]. A positive test for SARS-CoV-2 generally confirms the diagnosis of COVID-19, although false-positive tests are possible. If initial testing is negative but the suspicion for COVID-19 remains, the WHO recommends resampling and testing from multiple respiratory tract sites [34]. The accuracy and predictive values of SARS-CoV-2 testing have not been systematically evaluated. Negative RT-PCR tests on oropharyngeal swabs despite CT findings suggestive of viral pneumonia have been reported in some patients who ultimately tested positive for SARS-CoV-2. Serologic tests, once generally available, should be able to identify patients who have either current or previous infection but a negative PCR test [35,36]. Coinfection with SARS-CoV-2 and other respiratory viruses, including influenza, has been reported, and this may impact management decisions.

## 3. Management

There is no specific antiviral treatment recommended for COVID-19, and no vaccine is currently available [37]. The treatment is symptomatic, and oxygen therapy represents the major treatment intervention for patients with severe infection. Mechanical ventilation may be necessary in cases of respiratory failure refractory to oxygen therapy, whereas hemodynamic support is essential for managing septic shock [37]. Different strategies can be used depending on the severity of the patient and local epidemiology [38,39]. Home management is appropriate for asymptomatic or paucisintomatic patients. They need a daily assessment of body temperature, blood pressure, oxygen saturation and respiratory symptoms for about 14 days. Management of such patients should focus on prevention of transmission to others and monitoring for clinical status with prompt hospitalization if needed. Outpatients with COVID-19 should stay at home and try to separate themselves from other people in the household. They should wear a face mask when in the same room (or vehicle) as other people and when presenting to health care settings. Disinfection of frequently touched surfaces is also important. The optimal duration of home isolation is uncertain, but in consideration of incubation time around 14 days without symptoms (fever, dyspnoea, others) are considered sufficient to end home isolation.

Some patients with suspected or documented COVID-19 have severe disease that warrants hospital care. Management of such patients consists of ensuring appropriate infection control, and supportive care. Patients with severe disease often need oxygenation support. High-flow oxygen and noninvasive positive pressure ventilation have been used. Some patients may develop acute respiratory distress syndrome and warrant intubation with mechanical ventilation; extracorporeal membrane oxygenation may be indicated in patients with refractory hypoxia.

The main pharmacological experimental options are summarised in Table 2. Glucocorticoids should not be used in patients with COVID-19 pneumonia unless there are other indications (e.g., exacerbation of chronic obstructive pulmonary disease) [40,41]. Glucocorticoids have been associated with an increased risk for mortality in patients with influenza and delayed viral clearance in patients with Middle East respiratory syndrome coronavirus (MERS-CoV) infection. Although they were widely used in management of severe acute respiratory syndrome (SARS), there was no good evidence for benefit, and there was persuasive evidence of adverse short- and long-term harm [42,43].

A number of investigational agents are being explored for antiviral treatment of COVID-19, and enrolment in clinical trials should be discussed with patients or their proxies. Certain investigational agents have been described in observational studies or are being used anecdotally based on in vitro or extrapolated evidence. It is important to emphasize that there are no controlled data supporting the use of any of these agents, and their efficacy for COVID-19 is unknown.

Remdesivir is a novel nucleotide analogue that has activity against SARS-CoV-2 in vitro and related coronaviruses (including SARS and MERS-CoV) both in vitro and in animal studies. The compassionate use of remdesivir through an investigational new drug application has been described in various studies [44,45]. Any clinical impact of remdesivir on COVID-19 remains unknown.

Chloroquine and hydroxychloroquine have antiviral activity in vitro, as well as anti-inflammatory activities. They act on interference with the cellular receptor ACE2, on impairment of acidification of endosomes and on activity against many pro-inflammatory cytokines (e.g., IL-1 and IL-6) [46,47]. Other experiments have shown that azithromycin in combination with hydroxychloroquine appeared to have additional benefit, but there are methodologic concerns about the control groups for the study, and the biologic basis for using azithromycin in this setting is unclear [48].Despite the limited clinical data, given the relative safety of short-term use of hydroxychloroquine (with or without azithromycin), the lack of known effective interventions, and the in vitro antiviral activity, some clinicians think it is reasonable to use one or both of these agents in hospitalized patients with severe or risk for severe infection, particularly if they are not eligible for other clinical trials. The possibility of drug toxicity (including QT interval (QTc) prolongation and retinal toxicity) should be considered prior to using hydroxychloroquine, particularly in individuals who may be more susceptible to these effects including epilepsy, porphyria, myasthenia gravis, and retinal pathology—glucose-6-phosphate dehydrogenase (G6PD) deficiency [47,49].

Tocilizumab is a recombinant humanized monoclonal antibody which binds to the interleukin-6 (IL-6) receptor and blocks it from functioning. It is used for patients with severe COVID-19 and elevated IL-6 levels; the agent is being evaluated in a clinical trial [50].

Lopinavir-ritonavir appears to have little to no role in the treatment of SARS-CoV-2 infection [51]. This combined protease inhibitor, which has primarily been used for HIV infection, has in vitro activity against the SARS-CoV [36] and appears to have some activity against MERS-CoV in animal studies [52]. However, there was no difference in time to clinical improvement or mortality at 28 days in a randomized trial of 199 patients with severe COVID-19 given lopinavir-ritonavir (400/100 mg) twice daily for 14 days in addition to standard care versus those who received standard of care alone [53]. Moreover, limited evidence are available for baraticinib, a numb-associated kinase (NAK) inhibitor, with a particularly high affinity for the kinase AAK1, a pivotal regulator of clathrin-mediated endocytosis, anakinra, an anti IL-1, used in some UTI settings in Lombardy, Italy, and faviparavir a RNA-dependent RNA-polymerase inhibitor.

Support oxygen therapy with high-flow nasal oxygen (HFNO) should be used only in selected patients with hypoxemic respiratory failure. Compared with standard oxygen therapy, HFNO reduces the need for intubation. Patients with hypercapnia, hemodynamic instability, multiorgan failure, or abnormal mental status should generally not receive HFNO, although emerging data suggest that HFNO may be safe in patients with mild-moderate and non-worsening hypercapnia. Non-invasive ventilation (NIV) patients treated with either HFNO or NIV should be closely monitored for clinical deterioration. Mechanical ventilation is the main supportive treatment for critically ill patients [54,55].

In positive patients with a D-Dimer value four times higher than the normal limit, and without anticoagulant contraindications, an anticoagulation therapy is recommended [56]. The French Medicines Agency on its official page, warns of the possible harmful effects of nonsteroidal anti-inflammatory drugs (NSAIDs). The European Medicines Agency (EMA), for its part, undertakes to carry out an investigation in this regard and collect data but is not reluctant to advise against its use, however, it is advisable to prudently take paracetamol in the first instance [57].

Interestingly, there is hypothesis of the link between angiotensin converting enzyme (ACE) inhibitors and COVID-19. Indeed, SARS-CoV-2 uses ACE receptor 2 for entry into target and in animal experiments both lisinopril and losartan can significantly increase mRNA expression of cardiac ACE2. If this were the case, we might be able to reduce the risk of fatal COVID-19 courses in many patients by temporarily replacing these drugs [58]. Existing literature strongly recommends that healthy patients continue therapy, and in hospitalized patients to modify ACE-I/ARB with other therapy (calcium channel blockers) [59].

### 3.1. Prevention

Prevention is, so far, the best practice in order to reduce the impact of COVID-19 considering the lack of effective treatment. At the moment, there is no vaccine available and the best prevention is to avoid exposure to the virus [60]. In order to achieve this goal, the main measures are the following: (1) to use face masks; (2) to cover coughs and sneezes with tissues; (3) to wash hands regularly with soap or disinfection with hand sanitiser containing at least 60% alcohol; (4) to avoid contact with infected people; (5) to maintain an appropriate distance from people; and (6) to refrain from touching eyes, nose, and mouth with unwashed hands [61] (Table 3). Interestingly, the WHO issued detailed guidelines including: (I) Regularly and thoroughly clean your hands with an alcohol-based hand rub or wash them with soap and water; (II) Avoid touching eyes, nose and mouth; (III) Practice respiratory hygiene covering your mouth and nose with your bent elbow or tissue when you cough or sneeze; (IV) If you have fever, cough and difficulty breathing, seek medical care early; (V) Stay informed and follow advice given by your healthcare provider; (VI) Maintain at least 1 m (3 feet) distance between yourself and anyone who is coughing or sneezing [62]. In particular, regarding the use of face mask, health care workers are recommended to use particulate respirators such as those certified N95 or Filtering FacePiece 2 (FFP2) when performing aerosol-generating procedures and to use medical masks while providing any care to suspected or confirmed cases [63]. Moreover, while an individual without respiratory symptoms is not required to wear a medical mask when in public, people with respiratory symptoms are advised to use medical masks both in health care and home care settings [64].

### 3.2. Future Perspective

The COVID-19 outbreak is proving to be an unprecedented disaster, especially in the most afflicted countries including China, Italy, Iran and USA in all aspects, especially health, social and economic. It is too early to forecast any realistic scenario, but it will have a strong impact worldwide. If high income countries, especially those already affected by the outbreak, seem to face a catastrophic perspective, in low-income countries there seem to be two possible scenarios. In particular, in the worst-case scenario, when the COVID-19 outbreaks, the majority of countries will be unprepared, with low resources allocated for affording the viral emergency and the consequences will be catastrophic. In the best case scenario, similarly to the global outbreak of the SARS-CoV in 2003, also the COVID-19 will not affect Africa or South America on a large scale suggesting that respiratory viruses spread more effectively in the winter and, therefore, the southern hemisphere will be affected later in the year, if at all [63]. To this could contribute also the climate-specific cultural differences (living more outdoors than indoors), the effect of UV light on the survival of the virus on surfaces, immunological differences of the population (innate immunity), preexposure with coronaviruses, or the higher temperatures [64]. This data was also indirectly supported by Chin and colleagues that artificially reproduced different environmental conditions in order to study the virus survival capacity [65]. In addition to this hopeful low impact, if the prevention measures will be implemented, we could register a lower incidence of hygiene-linked diseases that still represent leading causes of death [66].

## 4. Conclusions

This review provides an insight into the COVID-19 current situation and represents a picture of the current state of the art in terms of public health impact, pathophysiology and clinical manifestations, diagnosis, case management, emergency response and preparedness. There is a rapidly growing body of literature on this topic and hopefully it will help in finding an effective vaccine and the best practice for the management and treatment of symptomatic cases. Only once this pandemic ends, one will be able to assess the health, social and economic impact of this global disaster and we should be able to learn lessons especially in terms of public and global health for any future similar pandemics.

## Figures and Tables

**Table 1 ijerph-17-02690-t001:** Main COVID-19-associated symptoms.

Fever
Cough
Dyspnea
Headach
Sore throat
Rhinorrhea

**Table 2 ijerph-17-02690-t002:** Main Corona Virus Disease 2019 (COVID-19) pharmacological experimental options.

Glucocorticoids
Remdesivir
Chloroquine and hydroxychloroquine *
Tocilizumab
Lopinavir-ritonavir
Baraticinib
Non-steroidal anti-inflammatory drugs
Angiotensin converting enzyme 2

* mainly in combination with **azithromycin**.

**Table 3 ijerph-17-02690-t003:** Main Corona Virus Disease 2019 (COVID-19) prevention measures.

To use face masks
To cover coughs and sneezes
To wash hands regularly
To avoid contact with infected people
To maintain an appropriate distance from people
To refrain from touching eyes, nose, and mouth
In case of symptoms, seek medical care early
To follow advice given by your healthcare provider

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
