# Peer review of "Coronavirus Diseases (COVID-19) Current Status and Future Perspectives: A Narrative Review"

_ijerph, 2020, doi:10.3390/ijerph17082690_

Round 1

Reviewer 1 Report

The present manuscript is very well organized and is of great interest for the current moment of Public Health in the world. I suggest a formatting review to the authors, especially on page 6, for later publication.

Author Response

Reviewer 1

The present manuscript is very well organized and is of great interest for the current moment of Public Health in the world. I suggest a formatting review to the authors, especially on page 6, for later publication.

Thank you, the formatting was done as suggested.

Reviewer 2 Report

This is a very interesting and informative review of the latest data from published literature related to COVID-19. The following suggestions are offered in order to strengthen the paper.

Title describes the work.

Abstract - the last sentence should explain what "state of the art" is being discussed. Public health, clinical treatment, emergency response and preparedness? Further explanation would motivate the reader.

Key words - COVID-19 is not a yet a MeSH term, but will likely become one. Therefore, keep this. Consider adding 'pathogenesis,' 'preparedness, emergency,' and 'pandemic' as these terms would improve the manuscript's searchability. Omit 'review.'

Tables would aid readability. Consider summary tables for each major section. Inclusion of search terms would be appropriate.

Page 3, paragraph 2 - spell out ACE2 (angiotensin converting enzyme 2). This is very important for understanding the mechanism of damage in target organs.

Page 6, paragraph 1 - The following sentence is copied verbatim from reference 57: 'Anticoagulation therapy is recommended for COVID-19 patients when the D-Dimer value is 4 times higher than the normal upper limit, except for patients with anticoagulant contraindications.[Dalteparin?] 100U per kg weight per 12 hours by subcutaneous injection for 3-5 days [is recommended as prophylaxis for venous thromboembolism].' It needs to be rephrased. The dosing recommendation is not particularly helpful, even from the reference. They are likely referring to dalteparin, since it is the only LMWH dosed in units. I included it in the sentence suggestion. The abbreviation, HFNO (high flow nasal oxygen), should be included in parenthesis after its first use in the manuscript.

Conclusion - again please expand on the phrase "state of the art."

References are not in mdpi style and need to be reformatted.

Author Response

Reviewer 2

Abstract - the last sentence should explain what "state of the art" is being discussed. Public health, clinical treatment, emergency response and preparedness? Further explanation would motivate the reader.

Thank you, as suggested we better specified as follow: “In particular, we focus on public health impact, pathophysiology and clinical manifestations, diagnosis, case management, emergency response and preparedness.”

Key words - COVID-19 is not a yet a MeSH term, but will likely become one. Therefore, keep this. Consider adding 'pathogenesis,' 'preparedness, emergency,' and 'pandemic' as these terms would improve the manuscript's searchability. Omit ‘review.'

Thank you, as suggested we removed “review” and we added “pathogenesis”, “preparedness”, “emergency”, and “pandemic”.

Tables would aid readability. Consider summary tables for each major section. Inclusion of search terms would be appropriate.

Thank you for your suggestion, we agree and we added the following three tables:

Table 1. Main COVID-19 associate symptoms

Table 2. Main COVID-19 pharmacological experimental options

Table 3. Main COVID-19 prevention measures

Page 3, paragraph 2 - spell out ACE2 (angiotensin converting enzyme 2). This is very important for understanding the mechanism of damage in target organs.

Thank you, as suggested we spelled out.

Page 6, paragraph 1 - The following sentence is copied verbatim from reference 57: 'Anticoagulation therapy is recommended for COVID-19 patients when the D-Dimer value is 4 times higher than the normal upper limit, except for patients with anticoagulant contraindications.[Dalteparin?] 100U per kg weight per 12 hours by subcutaneous injection for 3-5 days [is recommended as prophylaxis for venous thromboembolism].' It needs to be rephrased. The dosing recommendation is not particularly helpful, even from the reference. They are likely referring to dalteparin, since it is the only LMWH dosed in units. I included it in the sentence suggestion. The abbreviation, HFNO (high flow nasal oxygen), should be included in parenthesis after its first use in the manuscript.

Thank you, we rephrased as follow: “In positive patients with a D-Dimer value 4 times higher than the normal limit, and without anticoagulant contraindications, an anticoagulation therapy is recommended”. 

Moreover, we spelled out HFNO the first time.

Conclusion - again please expand on the phrase "state of the art.”

Thank you, as suggested we rephrased specifying as follow: This review provides an insight into  the COVID-19 current situation and represents a picture of the current state of the art in terms of public health impact, pathophysiology and clinical manifestations, diagnosis, case management, emergency response and preparedness.

References are not in mdpi style and need to be reformatted.

Thank you, we reformatted following instructions for authors.

Reviewer 3 Report

Comments

This mini-review about the COVID-19 described the epidemiology, pathophysiology, clinical manifestation, diagnosis, management and future perspective, is helpful and will attract interest from readers. In my opinion, this manuscript can be accepted after minor revision.

The major problems and suggestions are listed as follows:

(1) The data of epidemiology need further update. For example, Page 1 (“on March 11th 2020 the number of countries involved was 114, with more than 118,000 cases and over 4,000 deaths, WHO declared the pandemic status”) and Page 2 (“Today, 31st of march 2020, based on the WHO reports, we have globally 693,224 confirmed cases and 33,106 deaths distributed as follow: Western Pacific Region103,775 cases and 3,649 deaths, European Region 392.757 cases and 29.962 deaths, South East Asia Region 4,084 cases and 158 deaths, Eastern mediterranean Region 46,329 cases and 2,813 deaths, Region of the Americas 142,081 cases and 2,457 deaths and, Africa region 3,486 cases and 60 deaths[13]”)

(2) In the Abstract section, “Whuan” is spelled mistakenly, should be “Wuhan”

(3) In Introduction section, the outbreak time of SARS is not “November 2003”.

(4) “as well as significant community transmission occurring in several Asian countries including Iran and Italy and was declared by the World Health Organization (WHO) as global pandemic on the 11th March 2020”. This sentence creates ambiguity, Italy is not an Asian country.

(5) The style of reference needs strict revision.

(6) Some improper formats exist, see Page 6.

Author Response

Reviewer 3

  1. The data of epidemiology need further update. For example, Page 1 (“on March 11th 2020 the number of countries involved was 114, with more than 118,000 cases and over 4,000 deaths, WHO declared the pandemic status”) and Page 2 (“Today, 31st of march 2020, based on the WHO reports, we have globally 693,224 confirmed cases and 33,106 deaths distributed as follow: Western Pacific Region103,775 cases and 3,649 deaths, European Region 392.757 cases and 29.962 deaths, South East Asia Region 4,084 cases and 158 deaths, Eastern mediterranean Region 46,329 cases and 2,813 deaths, Region of the Americas 142,081 cases and 2,457 deaths and, Africa region 3,486 cases and 60 deaths[13]”)

Thank you for your suggestion. In our opinion some significant data as in page 1 are necessary. The 31st of march is the data of submission so the “most updated data”. We can further update in the last date before publishing.

(2) In the Abstract section, “Whuan” is spelled mistakenly, should be “Wuhan”

Thank you, as suggested we corrected.

(3) In Introduction section, the outbreak time of SARS is not “November 2003”.

Thank you, as suggested we corrected.

(4) “as well as significant community transmission occurring in several Asian countries including Iran and Italy and was declared by the World Health Organization (WHO) as global pandemic on the 11th March 2020”. This sentence creates ambiguity, Italy is not an Asian country.

Thank you, as suggested we rephrased the sentence changing “asian” by “worldwide”.

(5) The style of reference needs strict revision.

Thank you, we reformatted following instructions for authors.

(6) Some improper formats exist, see Page 6.

Thank you, the formatting was done as suggested.

Reviewer 4 Report

Overall, for perspective piece its very wordy and hard to follow.
It doesn’t provide any clear 'Future Perspective" as suggested in title,
so should be revised.
Also 'Abstract ' does not describe the key points of this article. Authors should pick main attributes of the disease-symptoms- treatment
options-disease management.

It would be also nice to include basic flow chart or
diagram summarizing the perspective.

I have attached my specific comments in the attached pdf.

Specific comments:

pathogenetic mechanism 

relevance of this point of glucocorticoids is not clear. Authors didnt mention any reference where they have used for COVID-19 patients

In vitro: italicized.

Its hard to comment about temperature sensitivity of the virus, however if authors make this suggestion they support with proper literature search. There are few articles as preprint (Medrxiv-Carleton and Meng-Casual emperical estimation..) and Biorxiv suggesting that Southern hemishpere may now start seeing uptick in cases but also will depend upon proper containment methods.

Authors should add relevant refereces, there is no real affect of UV light on virus. However Lancet article-Chin et al.-DOI:https://doi.org/10.1016/S2666-5247(20)30003-3 ) had detailed study on proper disinfectants and temperature sensitivity of the virus. 

Author Response

Reviewer 4

It doesn’t provide any clear 'Future Perspective" as suggested in title, so should be revised. 

Thank you for your comment. However we do not fully agree. As we have written, “It is too early to forecast any realistic scenario” and, again, we have hypnotised 2 different and possible scenario for LMIC. So in our opinion it makes sense to consider these “future perspectives”.

Also 'Abstract ' does not describe the key points of this article. Authors should pick main attributes of the disease-symptoms- treatment  options-disease management. 

Thank you, as suggested we better specified as follow: “In particular, we focus on public health impact, pathophysiology and clinical manifestations, diagnosis, case management, emergency response and preparedness.”

It would be also nice to include basic flow chart or  diagram summarizing the perspective.

Thank you for your comment. considering the few information available we think it is not fully appropriate to create a flow chart because it would be very poor and wouldn’t help to understand more than the text.

Specific comments

  1. pathogenetic mechanism 

Sorry, we do not understand why you are asking to spell.

  1. relevance of this point of glucocorticoids is not clear. Authors didnt mention any reference where they have used for COVID-19 patients

Sorry but we do not agree. References from 40 to 43 are specifically related to this issue. The fact that the utilisation of steroids is not clear reflect the contrasting data existing. We also performed a meta-analysis titled “USE OF CORTICOSTEROIDS IN CORONAVIRUS DISEASE 2019 PNEUMONIA: A SYSTEMATIC REVIEW OF THE LITERATURE” concluding that “The literature to date does not fully support the routine use of corticosteroids in COVID-19, but some findings suggest that methylprednisolone could lower mortality rate in more severe forms of the condition. Findings from ongoing and future intervention studies are needed to better understand the role of corticosteroids in the treatment of COVID-19”.

  1. In vitro: italicized.

Sorry but “in vitro” expression is commonly used in English.

  1. Its hard to comment about temperature sensitivity of the virus, however if authors make this suggestion they support with proper literature search. There are few articles as preprint (Medrxiv-Carleton and Meng-Casual emperical estimation..) and Biorxiv suggesting that Southern hemishpere may now start seeing uptick in cases but also will depend upon proper containment methods.

We agree. It is hard now to say what it will happen. About the temperature, we based our sentence also on this study: Hopman J, Allegranzi B, Mehtar S. Managing COVID-19 in Low- and Middle-Income Countries. JAMA. 2020 Mar 16. doi: 10.1001/jama.2020.4169

  1. Authors should add relevant refereces, there is no real affect of UV light on virus. However Lancet article-Chin et al.-DOI:https://doi.org/10.1016/S2666-5247(20)30003-3 ) had detailed study on proper disinfectants and temperature sensitivity of the virus.

Thank you for your suggestion. Honestly, we think JAMA is quite relevant as reference. However, we also add the suggested reference.